# Red Sea Atlas of Coral-Associated Bacteria Highlights Common Microbiome Members and Their Distribution across Environmental Gradients—A Systematic Review

**DOI:** 10.3390/microorganisms10122340

**Published:** 2022-11-26

**Authors:** Nathalia Delgadillo-Ordoñez, Inês Raimundo, Adam R. Barno, Eslam O. Osman, Helena Villela, Morgan Bennett-Smith, Christian R. Voolstra, Francesca Benzoni, Raquel S. Peixoto

**Affiliations:** 1Marine Microbiomes Laboratory, Red Sea Research Center (RSRC), Biological and Environmental Sciences and Engineering Division (BESE), King Abdullah University of Science and Technology (KAUST), Thuwal 23955, Saudi Arabia; 2Department of Biology, University of Konstanz, 78457 Konstanz, Germany

**Keywords:** coral reefs, Red Sea, microbiology, microbial ecology, bacterial diversity, coral-associated bacteria

## Abstract

The Red Sea is a suitable model for studying coral reefs under climate change due to its strong environmental gradient that provides a window into future global warming scenarios. For instance, corals in the southern Red Sea thrive at temperatures predicted to occur at the end of the century in other biogeographic regions. Corals in the Red Sea thrive under contrasting thermal and environmental regimes along their latitudinal gradient. Because microbial communities associated with corals contribute to host physiology, we conducted a systematic review of the known diversity of Red Sea coral-associated bacteria, considering geographic location and host species. Our assessment comprises 54 studies of 67 coral host species employing cultivation-dependent and cultivation-independent techniques. Most studies have been conducted in the central and northern Red Sea, while the southern and western regions remain largely unexplored. Our data also show that, despite the high diversity of corals in the Red Sea, the most studied corals were *Pocillopora verrucosa*, *Dipsastraea* spp., *Pleuractis granulosa*, and *Stylophora pistillata*. Microbial diversity was dominated by bacteria from the class Gammaproteobacteria, while the most frequently occurring bacterial families included *Rhodobacteraceae* and *Vibrionaceae*. We also identified bacterial families exclusively associated with each of the studied coral orders: Scleractinia (*n* = 125), Alcyonacea (*n* = 7), and Capitata (*n* = 2). This review encompasses 20 years of research in the Red Sea, providing a baseline compendium for coral-associated bacterial diversity.

## 1. Introduction

Corals are cnidarian hosts that live in association with several groups of microorganisms, including viruses, bacteria, archaea, dinoflagellate algae, and fungi, which together comprise the so-called holobiont [1,2]. The endosymbiotic relationship between the coral host and photosynthetic dinoflagellates (family *Symbiodiniaceae*) provides up to 90% of the host nutritional demands, and it is critical to sustaining reef ecosystem functioning [3,4]. In addition to *Symbiodiniaceae*, other microorganisms, notably prokaryotes (referred to as the microbiome hereafter), associate with corals and play an important role in coral physiology and health [5,6,7,8,9,10]. The coral microbiome is involved, among other chemical and biological processes, in carbon, nitrogen, and sulfur cycling, defense against pathogens, environmental adaptability, and degradation of toxic compounds [9,10,11,12,13,14,15,16,17]. However, the elaborated interactions between the host and its microbiome are complex, and it is yet to be fully understood how microbial communities can improve coral holobiont functions to tolerate environmental stressors, particularly those derived from ocean warming [9,13].

Coral-associated microbiomes are dynamic communities that show flexibility (e.g., restructuring of the microbial community) to changing environmental conditions as a potential adaptive mechanism for holobiont health [18,19]. Overall, these microbial communities present (i) host specificity, where each coral species has, to some extent, distinct associated microbial groups that compose species-specific microbiomes [20,21]; (ii) variable microbial groups that respond to environmental conditions [20]; and (iii) distinct communities between coral compartments, i.e., mucus, tissue, and skeleton [22,23]. As such, coral hosts are likely to select the most advantageous microbial partners that contribute to adaptation to environmental perturbations and, consequently, their survival [24,25]. Therefore, understanding the composition, diversity, and dynamics of microbiomes associated with corals across environmental settings is key to understanding the potential role of the microbiome in improving coral acclimation to environmental stressors, including those derived from ongoing ocean warming.

The Red Sea is a biodiversity hotspot that harbors thousands of marine species and is populated by extensive and healthy coral reefs, with a high rate of endemism [26,27]. Coral holobionts in the Red Sea have adapted to an overall unique marine environment characterized by a combination of conditions deemed extreme both in shallow and deep water compared to other regions. In the shallow photic zone, Red Sea corals experience salinity of 41 PSU and summer temperatures range from 25–32 °C along the latitudinal gradient [28,29]. In the aphotic zone, the water temperature never drops below 21 °C and salinity remains around 40.5 PSU. Similarly, the chlorophyll concentration is higher in the southern Red Sea due to the exchange with nutrient-rich seawater from the Gulf of Aden, and consequently, seawater visibility in the south is lower than in the central and northern Red Sea regions [30]. Despite the contrasting environmental conditions, genetic analysis of coral species have shown high gene flow and genetic similarity between coral populations along the latitudinal gradient of the Red Sea, suggesting that coral hosts have low genetic variations along the Red Sea latitudinal gradient [31,32]. Conversely, patterns of continuous latitudinal zonation have been highlighted in selected scleractinian coral hosts for the associated *Symbiodiniaceae* communities [33,34]. All of the above may highlight the potential of microbiome restructuring as an adaptive strategy to influence species distribution along the Red Sea to likely facilitate survival of the same coral species under adverse environmental regimes. Osman and colleagues (2020) found plasticity of microbiome communities across latitudinal gradients of the northern Red Sea, but high specificity of endosymbiotic algae [21]. As such, the Red Sea can be seen as a natural laboratory to assess the response of coral microbiomes to different environmental conditions and how microbiome flexibility could increase the coral holobiont tolerance to environmental shifts.

Despite the increasing number of microbiome studies in recent years, e.g., [10,35], the number of publications on the Red Sea remains relatively small in comparison to other regions, such as the Great Barrier Reef, and has focused only on a few coral species and/or has been restricted to certain regions [27]. Few studies have assessed the coral microbiome composition and its dynamics along its latitudinal gradient in the Red Sea, e.g., [21,36,37]. Here, we performed a systematic review of the current knowledge of the Red Sea coral associated bacteria using publicly accessible data. The objectives were: (1), to assess and summarize the existing Red Sea coral bacteria information published over the past two decades and (2), to detect current knowledge gaps to inform the direction of future fundamental and applied research. Our analysis of the published data allowed us to assess the bacterial composition of several Red Sea coral taxa. Moreover, we identify bacterial taxa associated with healthy corals and their distribution patterns along the latitudinal gradient of the Red Sea. Thus, we provide the first comprehensive atlas of coral-associated bacteria in the Red Sea, serving as a baseline for future fundamental and applied research including coral restoration projects.

## 2. Materials and Methods

### 2.1. Red Sea Studies Survey

A systematic survey for published Red Sea coral-associated bacteria was conducted according to the PRISMA guidelines [38], using the Web of Science and PubMed public databases from July 2002 until February 2021 (https://www.webofscience.com/wos/woscc/basic-search (accessed on 1 February 2021); https://pubmed.ncbi.nlm.nih.gov/ (accessed on 1 February 2021), respectively). The available literature published over the past two decades along the Red Sea was compiled using five keywords for the web search: “Red Sea”, “coral”, “bacteria”, “microbiome” and “microbes”, in their different combinations, using the restrictor “AND” to limit the search. The search produced 151 results, and publications that were not from the Red Sea were excluded. The retrieved papers were downloaded and each paper was assessed to determine whether it was relevant to the coral associated bacteria or not, excluding the publications that studied free-living marine bacteria, marine sediments, aquaculture, and marine environments different from coral reefs. Thereafter, a total of 54 studies were selected, and metadata for each paper were extracted, including the year of publication, the studied coral species, geographic location (coordinates), sampling depth, sequencing technology (i.e., culture-based vs. sequencing), and associated GenBank accession numbers. The publications and studied coral species were counted to quantify the number of coral microbiome studies along the latitudinal gradient of the Red Sea. The geographic coordinates of the sampling locations were retrieved from each study. However, for the studies that did not provide this information (*n* = 9), the coordinates of the sampling sites were estimated using Google Earth based on the available information in each reference. For this, coordinates were estimated by reading the location description section in the retrieved paper to locate the sampling site on Google Earth. Then, a middle point was taken as a representative coordinate for the study. Coordinates were transformed into a decimal system and plotted over the Red Sea map using ArcGIS Pro software (V 2.8.0). The Fishnet tool in ArcGIS pro (V 2.8.0) was used to quantify the number of studied sites per 100 km^2^ grid in the different regions of the Red Sea.

### 2.2. Coral-Associated Bacteria in the Red Sea

We used the data from the retrieved publications to obtain a list of bacteria associated with different coral species along the Red Sea. We focused on bacterial communities associated with healthy corals (*n* = 36 publications) because our objective was an assessment of the coral associated bacterial communities and their variability under natural conditions in the study area. Thus, samples detailing diseased, bleached, temperature- or nutrient-treated coral and seawater samples were removed from the analysis (*n* = 18 publications, see S1). Samples for which it was not possible to retrieve the data were similarly excluded. Furthermore, we only considered studies involving culture-independent techniques (*n* = 24 publications) for the bacterial atlas. Bacterial diversity retrieved from culture-dependent techniques was considered separately (*n* = 12 publications). The taxonomic profile for each sample was extracted from NCBI using the samples’ respective Sequence Read Archive (SRA) and/or accession numbers provided. Bacterial taxonomy was extracted from each unique SRA linked to 16S rRNA BioSamples using the Taxonomy Analysis tool in NCBI. The compiled taxonomic profiles for biological replicates per coral taxa within each study were aggregated and duplicate bacterial taxa were removed. Therefore, for each study, each bacterial taxon was represented only once, as presence/absence, for the studied corals. Here, we assessed the previously reported diversity of bacteria associated with corals, and did not compare and/or reanalyze the data. Therefore, in this atlas we included publications regardless of their different sequencing methods (Sanger vs. Next Generation Sequencing (NGS)), 16S rRNA gene regions (e.g., V2 vs. V3), platforms (e.g., HiSeq vs. MiSeq), pipelines (e.g., QIIME vs. mothur), and taxonomy resolution (OTUs = Operational Taxonomic Unit vs. ASVs = Amplicon Sequence Variants). The lowest taxonomic level that provided the most reliable information was the family level; hence, all the results presented were pooled to this taxonomic unit level of resolution. In addition, SRA annotations were missing in two publications [21,39]. In these cases, we used the taxonomic information provided within the publication.

The occurrence of each bacterial family was counted for all studied coral species along the Red Sea, excluding duplicates and biological replicates to avoid erroneous estimations in the data. Similarly, the reported diversity of coral bacterial communities between different Red Sea regions was assessed. For this, the Red Sea was subdivided into three regions known to have a different environmental regime [28,40,41], namely the northern, central, and southern Red Sea (lat. 29, 24, and 20, respectively). The number of reported bacterial families within each region was counted according to the available retrieved data from each publication to investigate the similarity and disparity between and among regions. In addition, we identified common bacterial families across coral species to highlight their possible roles in the holobiont in the context of future management directions and restoration applications.

The classification of the studied corals was updated to reflect the most recent taxonomic framework based on formal published revisions as per the World Register of Marine Species [42] and references therein. When outdated, the coral taxa names used in the original examined papers were modified. For example, for literature reports of *Favia favus* (Forskål, 1775), *Fungia granulosa* (Klunzinger, 1879), and *Fungia scutaria* (Lamarck, 1801), we adopted the most recently revised genus assignation, namely *Dipsastraea favus*, *Pleuractis granulosa*, and *Lobactis scutaria*, respectively. For examined samples from coral hosts lacking a deposited DNA and/or skeleton voucher, allowing re-identification of a potentially doubtful species-level assignation [43], the data were included in the analyses and *cf* was added to the species name to reflect such uncertainty. An abbreviation of *confer* in Latin, literally “compare with” (*cf*), is used in the taxonomic literature before a taxa name to indicate a level of uncertainty for its identification [44]. In our case, specifically, material identified as *Pocillopora damicornis* (Linnaeus, 1758) in the examined studies is here referred to as *P. cf damicornis* due to a lack of genetic identification of the examined material in the original paper. *Pocillopora* is a notoriously complex and diversified coral genus that poses particular challenges to in situ identification due to its colony plasticity and limited informative microstructural characteristics [45]. Lately, reliable species-level identification for this genus has, therefore, mostly been performed genetically. While *Pocillopora verrucosa* has been recorded as present in the Red Sea, e.g., [31], the presence of *P*. *damicornis* has not been confirmed thus far based on the published literature [46]; (Schmidt-Roach *pers*. *comm*.). Similarly, the fungiid *Ctenactis echinata* (Pallas, 1766) is common and locally abundant in the Red Sea [30,47,48]; however, the presence of its congener *Ctenactis crassa* (Dana, 1846) in the region, although reported in one of the references we examined here [49], has not yet been confirmed based on genetic data evidence. Therefore, while data from the study were still included in our analyses, the coral host is reported as *C. cf crassa* to reflect the aforementioned species level record uncertainty.

## 3. Results

### 3.1. Coral-Associated Bacteria Literature Survey

The assessment of the published literature retrieved on the Web of Science and PubMed databases revealed the presence of 54 studies on coral bacterial communities along the Red Sea over the past two decades (Table 1). These studies covered 152 sampling sites and the central Red Sea had the greatest number of sites (*n* = 85), followed by the north (*n* = 56), then south (*n* = 11) (Figure 1a). During 2000–2011, the total number of studies was 16 and this increased to 54 in the past decade (Figure 1b).

After retrieving the studies from the databases, and prior to excluding those not matching our criteria for the Red Sea bacterial atlas (see Figure 1d), the number of studied coral taxa was 67 (44 corals *sensu lato* classified to species and 23 to genus level). Among corals, broad term grouping representatives of multiple classes in the phylum Cnidaria, hard corals (*n* = 47, 70%), including scleractinians (Anthozoa, Hexacorallia, Scleractinia) and hydrocorals (Hydrozoa, Anthoathecata, Capitata), were the most represented, followed by soft corals and gorgonians (Anthozoa, Octocorallia, Alcyonacea) (*n* = 20, 30%). Out of 54 references, 76% (*n* = 41 papers) investigated the microbiome communities associated with healthy corals, whereas 17% were conducted on corals showing signs of disease (*n* = 9). Four studies focused on the microbial communities associated with seawater in coral reefs and/or microbial biofilms in artificial structures (Autonomous Reef Monitoring Structures “ARMS” or Terracotta Tiles) rather than on specific coral species (representing 7% of the total studies) (Figure 1c). Furthermore, culture-independent techniques were the most commonly used methods to identify the coral associated bacteria (*n* = 32 papers, 57%) compared to culture-dependent techniques, which were relatively less common (*n* = 20, 39%), while 3.7% of the studies (*n* = 2) used both culture-dependent and -independent approaches (Figure 1c). In addition, most of the studies focused on the photic zone (*n* = 52 papers, 96.3%) up to 50 m deep, whereas only 2 studies (4%) targeted deep-sea corals (aphotic zone) between 300 and 1000 m (Figure 1c). The predominant corals studied were *Pocillopora verrucosa* (Ellis and Solander, 1786) (*n* = 8 publications, 15%), followed by *Favia* sp. (= *Dipsastraea* sp.), *Pleuractis granulosa* (*n* = 7, 13%), *Stylophora pistillata* (Esper, 1792) (*n* = 6, 11%), *Acropora hemprichii* (Ehrenberg, 1834) (*n* = 4, 7%), *Dipsastraea favus* (Forskål, 1775), and *Platygyra* sp. (*n* = 4, 7%, each). Additional information from the retrieved studies is available in Appendix A.

### 3.2. Bacterial Diversity of Red Sea Corals from Cultivation-Independent Techniques

Out of 54 retrieved studies, the 24 that investigated coral-associated bacteria with healthy corals using culture-independent techniques were included in the diversity analysis/atlas (Figure 1d). Overall, we found 43 phyla, 67 classes, 150 orders, and 324 bacterial families associated with 35 taxa of healthy corals (Appendix A). Proteobacteria was the phylum with the highest occurrence across all coral taxa (i.e., 49.8%), followed by Firmicutes (11.9%), Actinobacteria (10.4%), Bacteroidetes (10%), Acidobacteria (3.1%), Verrucomicrobia (2%), Cyanobacteria (1.9%), and Planctomycetes (1.4%), which together comprised 90.7% of the total reported phyla across all studies (Figure 2a). Gammaproteobacteria (20.4%) and Alphaproteobacteria (17%) were the most identified classes within the phylum Proteobacteria, followed by Betaproteobacteria (5.5%), Deltaproteobacteria (4.8%), and Epsilonproteobacteria (1.2%) (Figure 2b). The occurrence of further phyla, classes, and orders is detailed in Figure 2a–c and Appendix A. The families with the highest occurrence across all coral taxa and sites were *Rhodobacteraceae* (1.4%—Alphaproteobacteria), *Pseudomonadaceae* (1.3%—Gammaproteobacteria), *Vibrionaceae* (1.3%—Gammaproteobacteria), *Flavobacteriaceae* (1.2%—Bacteroidetes), *Peptostreptococcaceae* (1%—Clostridia), *Burkholderiaceae* (1%—Betaproteobacteria), *Enterobacteriaceae* (0.9%—Gammaproteobacteria), *Moraxellaceae* (0.9%—Gammaproteobacteria), *Alteromonadaceae* (0.8%—Gammaproteobacteria), and *Xanthomonadaceae* (0.8%—Gammaproteobacteria), which together comprised 10.7% of all reported families (Figure 2d). Families representing less than 0.5% of the occurrences were groupedas “Others” and comprised 52.5% of the occurrences. Unclassified families represented 21.2% of all the reported families (Appendix A).

The presence/absence of each bacterial family within each coral host taxon was counted to assess their richness and provide a descriptive survey of their associations (Figure 3, Figure 4 and Figure 5). Seven coral taxa had a higher number of reported bacterial families (*n* > 130) (i.e., the hard corals *Stylophora pistillata*, *Seriatopora hystrix*, *Dipsastraea favus*, *Porites nodifera* and *Pocillopora cf damicornis*, and the soft corals *Sarcophyton trocheliophorum* and *Xenia umbellata*). The remaining coral species presented a moderate or low number of bacterial families identified by the available publications, ranging from 76 families in *Pocillopora verrucosa* to 2 families in *Acropora humilis* (Figure 3a). Interestingly, 52 distinct bacterial families were commonly reported in 10 or more different coral taxa (Figure 3b), while others were exclusively identified in some corals. For instance, the families *Rhodobacteraceae*, *Vibrionaceae*, *Pseudomonadaceae*, *Peptostreptococcaceae*, *Flavobacteriaceae*, *Burkholderiaceae*, and *Enterobacteriaceae* were reported in >60% of the corals studied, considering a range of sequencing technologies, depths, and bioinformatics analyses utilized per study, while 93 families have been described in unique coral taxa of these studies (Appendix A). A descriptive Atlas of our current knowledge on the bacterial diversity (represented here at the family level) reported for 32 coral species is shown in Figure 4, in addition to the distribution of the study sites in the publications used for the Atlas per coral species along the Red Sea.

### 3.3. Common and Unique Bacteria in Red Sea Corals

We found several bacterial families that exist in coral host taxa regardless of their region of provenance in the Red Sea. For example, *Rhodobacteraceae* was identified in 30 coral taxa out of the 35 studied (85.7%), followed by *Vibrionaceae* (*n* = 29, 82.9%), *Pseudomonadaceae* (*n* = 27, 77.1%), *Peptostreptococcaceae* (*n* = 23, 65.7%), and *Flavobacteriaceae* (*n* = 23, 65.7%). Furthermore, families such as *Burkholderiaceae* (*n* = 22, 62.9%), *Enterobacteriaceae* (*n* = 21, 60%), *Caulobacteraceae*, *Alteromonadaceae*, *Moraxellaceae* (*n* = 19, 54.3% each), and *Rhodospirillaceae* (*n* = 18, 51.4%) were found in more than 50% of the coral taxa, followed by *Xanthomonadaceae* (*n* = 17, 48.6%), *Comamonadaceae*, *Phyllobacteraceae* (*n* = 16, 45.7% each), *Staphylococcaceae*, *Alcaligenaceae*, *Sphingomonadaceae*, and *Streptococcaeae* (*n* = 15, 42.8% each), which were slightly less frequent. Other families, such as *Oxalobacteraceae*, *Propionibacteraceae*, *Chromatiaceae* (*n* = 14. 40% each), *Mycobacteriaceae*, *Streptomycetaceae*, *Rhizobiaceae*, *Lactobacillaceae*, *Corynebacteriaceae*, *Helicobacteraceae* (*n* = 13, 37.1% each), *Halomonadaceae*, *Clostridiaceae*, *Endozoicomonadaceae*, *Collwelliaceae*, *Hyphomonadaceae*, and *Dietziaceae* (*n* = 12, 34.3% each) were less commonly identified but still had a comparably high occurrence across coral taxa. Other frequently found bacteria are detailed in Figure 3b.

We also explored the diversity of bacterial families associated with the three coral orders (Scleractinia, Alcyonacea, and Capitata) to investigate common and unique families reported for each coral group (Figure 5a,b). We found 26 families that were commonly associated with the three coral orders, including some of the most frequently reported families (e.g., *Rhodobacteraceae*, *Vibrionaceae*, *Pseudomonadaceae*, *Peptostreptococcaceae*, and *Flavobacteraceae*). Interestingly, we also found exclusive bacterial associations for each of the three groups. For instance, 125 families were exclusively associated with Scleractinia. Among them, *Wekselliaceae* was the most frequently reported family across corals of this group (identified eight times, corresponding to 0.3% of the total microbial identifications), followed by *Yersiniaceae*, *Balneolaceae*, *Cellvibrionaceae*, *Chlorobiaceae*, *Methylothermaceae*, *Pseudonocardiaceae*, and *Saccharospirillaceae* (*n* = 4, 0.2% each). Alcyonacea presented a lower number of exclusive families (*n* = 7), such as *Coriobacteriaceae* (*n* = 3, 0.1%), *Elusimicrobiaceae*, *Rhodanobacteraceae*, *Cardiobacteriaceae*, *Atopobiaceae*, *Waddliaceae* and *Candidatus Midichloriaceae* (*n* = 1, 0.04% each), while only two bacterial families were exclusively associated with Capitata (*Neomegalonemataceae* and *Sphingosinicellaceae*, *n* = 1, 0.04%) (Figure 5a,b; Appendix A).

### 3.4. Coral Bacteria across Red Sea Regions

The assessment of bacterial communities among the Red Sea regions, based on the 24 studies assessed, revealed a decline in the number of bacterial families along the latitudinal gradient toward the south. The northern Red Sea had the highest number of families (*n* = 282), followed by the central (*n* = 203) and southern (*n* = 41) regions. The families with the highest occurrence in the northern region were *Rhodobacteraceae*, *Vibrionaceae*, and *Moraxellaceae* (i.e., 0.7% each, of the total occurrence for this region), followed by *Flavobacteriaceae* (0.68%), *Alteromonadaceae*, *Enterobacteriaceae*, *Pseudomonadaceae*, *Sphingomonadaceae*, and *Staphylococcaceae* (0.6% each). In the central Red Sea, the most frequently identified families were *Rhodobacteraceae*, *Pseudomonadaceae* (2.8% each), *Vibrionaceae* (2.7%), *Flavobacteriaceae* (2.4%), *Peptostreptococcaceae* (2.2%), and *Burkholderiaceae* (2.1%). Finally, the families with the highest occurrence in the southern region were *Endozoicomonadaceae* and *Rhodobacteraceae* (5.5% each), *Alcaligenaceae*, *Alteromonadaceae*, *Anaplasmataceae*, *Clostridiaceae*, and *Pseudomonadaceae* (3.7% each) (Appendix A).

Thirty-six families were common across these regions (e.g., *Rhodobacteraceae*, *Vibrionaceae*, *Flavobacteriaceae*, *Pseudomonadaceae*, and *Endozoicomonadaceae*, among others). Some of these families were also the most frequently identified across coral taxa (*Rhodobacteraceae*, *Vibrionaceae*, *Pseudomonadaceae*, and *Flavobacteriaceae*), as mentioned previously. Additionally, 115 families were exclusively reported in the northern region only, 40 families in the central region, and only 3 families were found exclusively in the southern Red Sea (*Thiotrichaceae*, *Acholeplasmataceae*, *Nocardiopsaceae*) (Figure 3c; Appendix A).

### 3.5. Cultured Bacterial Diversity of Red Sea Corals

We examined the bacterial diversity from 12 studies that used culture-dependent techniques to isolate bacteria from healthy corals (Appendix A). From a total of 21 coral taxa, including hard (*n* = 10 coral taxa, 47.6%) and soft corals (*n* = 11, 52.4%), the coral with the highest number of reported bacterial families was *Lobactis scutaria* (*n* = 12), followed by *Sarcophytum glaucum* (*n* = 9), *Acropora cytherea* (*n* = 8), *Pleuractis granulosa* (*n* = 6), *Stylophora* sp. (*n* = 5), *Xenia* sp. (*n* = 4), *Favia* sp., *Acropora hemprichii*, *Acropora humilis* (*n* = 4 each), *Platygyra* sp. (*n* = 3), *Porites* sp., *Pocillopora* sp. (*n* = 3 each), *Litophyton arboreum*, *Rhytisma fulvum*, *Lobularia* sp. (*n* = 2 each), and *Dendronephthya sinaiensis*, *Heteroxenia fuscescens*, *Dendronephthya hemprichi*, *Rhytisma fulvum fulvum*, *Sinularia* sp., and *Millepora dichotoma* (*n* = 1 each) (Figure 6a).

The reported bacterial diversity was considerably lower than in the studies that used culture-independent techniques. For instance, only 4 phyla, 5 classes, 22 orders, and 33 families were reported. The isolated bacterial diversity was composed of 47 isolates that belonged to the phylum Proteobacteria (approximately 61%), followed by Actinobacteria (24.6%) and Firmicutes (12.9%), while only one belonged to Bacteroidetes (1.3%). The classes were dominated by Gammaproteobacteria (46.7%), followed by Actinomycetia (24.6%), Alphaproteobacteria (14.3%), Bacilli (12.9%), and Cytophagia (1.3%). Orders were dominated by Alteromonadales (15.6%), Bacillales (12.9%) and Vibrionales (10.4%), followed by 18 other less-represented groups (Figure 6b; Appendix A). A total of 31 bacterial families (out of 33 reported families) were common to both culture-dependent and culture-independent techniques studies, and only two families (*Amorphaceae* and *Salinisphaeraceae*) were exclusively found in studies using culture-dependent techniques, isolated from *Pleuractis granulosa* and *Acropora cytherea*, respectively [16,53].

## 4. Discussion

In the past two decades, coral-associated microbes have been highlighted as key players in the functioning and resilience of the coral holobiont [7,35,96,97,98]. The rapid decline of coral reefs worldwide has escalated the need for new technologies and strategies to accelerate coral adaptation, including those based on microbial therapies [9,11,15,17,99,100,101]. Red Sea corals, especially in the northern Red Sea and the Gulf of Aqaba, are considered among the most resilient in the world, despite their exposure to “harsh conditions” such as high thermal anomalies, marked seasonality [28], and high salinity [102]. Northern Red Sea corals have a much higher thermal tolerance in relation to their prevailing mean maximum summer temperatures, although corals from the central Red Sea surpass these thresholds in absolute terms [81]. Red Sea corals, particularly from the central and southern regions, have presumably evolved strategies to survive these conditions (e.g., transcriptome plasticity) [82]; instead, the microbiome appears to contribute to the overall resilience, as some studies have found traits of the associated microbiome that likely contribute to coral survival, e.g., thermal tolerance, e.g., [21,64]. Understanding the coral-associated microbial composition provides a background for disentangling the complexity of microbiome–host interactions.

### 4.1. Red Sea Coral Microbiome Survey

This literature survey revealed that most of the reported studies in the past decade have focused on healthy corals [18,64], in contrast to previous years when studies predominantly targeted pathogenic bacteria related to coral diseases (as reviewed in Berumen et al., 2013) [103]. This trend reflects a change of direction in this research topic and contributes to increasing knowledge of the whole coral microbiome and an evaluation of its importance for the health and resilience of the holobiont. However, it might also be related to the fact that, compared to other areas, coral disease prevalence is still low in the Red Sea [104]. While the studies in the first decade of the past 20 years were focused on describing the microbiome of in situ collected corals, e.g., [64], those in the second half focused on microbiome dynamics as a result of experimental manipulation [81,82]. Another interesting aspect is that the studies are mostly aggregated in certain parts of the Red Sea; the majority of sampling efforts have been carried out in the north and central regions along the Saudi Arabian Red Sea coast, leaving a vast territory unexplored. In addition, the studies in the northern Red Sea have mostly been performed in the Gulf of Aqaba. In contrast, the southern and western regions show a gap in microbiome research, with only a few publications available (see Table 1). This lack of information was reflected in the reported bacterial diversity throughout the Red Sea: only 41 bacterial families were reported to be associated with corals in the south, unlike the northern and central regions with a remarkably higher number of reported families (*n* = 282 and 204, respectively). This highlights the urgent need to expand microbiome research in unexplored areas along the Red Sea coastline.

Despite the high diversity of corals found in the Red Sea (*n* = 360 coral species) [27,105], microbiome communities were examined in only 67 coral taxa (representing approximately 18% of the total described coral diversity in the Red Sea), highlighting the need for further microbiome studies that comprehensively cover this vast coral diversity. The most commonly studied corals (e.g., *Pocillopora verrucosa*, *Dipsastraea* sp., *Pleuractis granulosa*, and *Stylophora pistillata*) are hard corals (subclass: Scleractinia) and are widely distributed in the Red Sea [48]. The reasons for the selection of these species are difficult to define; it may vary from a random selection to higher abundance in specific sampling sites or the ability of each specific team to identify and access these species, among other potential explanations. In addition, Scleractinian corals are commonly found in mid-shallow waters (i.e., 0–30 m depth), making them good candidates for coral microbiome studies, as they are relatively easy to spot and manipulate in ex situ conditions [64,67,81,82]. Additionally, in the Red Sea, scleractinian coral diversity is higher than soft coral diversity in mid-shallow waters [106,107]. In our survey, soft corals were less represented compared to hard corals (the number of studies per coral taxa ranged from 1–3 in comparison to 1–8 studies for soft and hard corals, respectively). More efforts to explore soft-coral diversity would contribute not only to discovering microbiome–host interactions but also to unlocking their biotechnological potential since some studies have found unique chemical and microbial traits with promising applications, e.g., bioactive natural products (e.g., [108,109,110,111,112,113,114]). In the Red Sea, the vast majority of studies on the diversity, biology, and taxonomy of soft corals have been performed in the Gulf of Aqaba [115,116,117,118]. However, the soft coral fauna remains poorly studied in the rest of the Red Sea in comparison to other places, such as the Great Barrier Reef in Australia or the Caribbean Sea [108,119].

Another aspect to be considered is that most of the corals reported here were studied in mid-shallow waters. In comparison, only 3 deep-sea coral species (*Eguchipsammia fistula*, *Dendrophyllia* sp., and *Rhizotrochus typus*) have been studied [70,71], highlighting the importance of exploring microbes associated with deep-sea corals in the Red Sea [70,120,121,122,123], which are overall poorly understood and have been shown to possess physiological and metabolic plasticity [124]. This plasticity is hypothesized to be strongly linked to their microbiome, which presumably facilitates survival in the deep-sea environment [70,71,79,122,124,125,126]. For example, recent studies have reported the presence of bacterial taxa that can confer putative beneficial roles (e.g., those involved in the nitrogen and carbon cycles) associated with the deep Red Sea corals *E. fistula*, *Dendrophyllia* sp. and *R. typus* [71,127]. As such, unveiling the specific microbiome roles in their adaptation is highly relevant to understanding microbiome dynamics in such challenging conditions and may reveal further functions for future applications.

Additionally, as previously mentioned, the Red Sea has a high coral endemism [30], but studies on endemic species are still in their infancy. We found no related publication of the bacterial communities of Red Sea endemic corals, which highlights the opportunity to study endemic corals that potentially possess differential traits. Such specific traits and potential unique bacterial partners could be particularly interesting if associated with the adaptive capacity of endemic corals along environmental gradients of the Red Sea [19,34,36] through microbiome flexibility and plasticity [19].

Overall, new studies focusing on keystone coral species that are abundant and/or endemic in the Red Sea and largely contribute to the coral reef landscape could be extremely beneficial to contribute to conservation strategies, for example, by exploring the native associated microorganisms that play an active role in their state of health [81,82,98].

### 4.2. Culture-Independent Bacterial Diversity

Bacterial families are widely distributed across different corals, Red Sea regions, and studies. For instance, the families *Vibrionaceae* and *Rhodobacteraceae* were found in 29 and 30 corals, respectively, of the 35 corals studied, which suggests that they are ubiquitous in the microbiome of Red Sea corals. Bacterial members of the family *Vibrionaceae* are extensively present in the marine environment and are known for their role as opportunistic or pathogenic bacteria [16,86,88,128,129]. However, members of the *Vibrionaceae* family can also play a role in the cycling of nutrients [130] by taking up dissolved organic matter. It has also been reported that *Vibrios* can provide essential polyunsaturated fatty acids, produce enzymes to degrade chitin (the second most abundant polymer and main source of amino sugars in the oceans [131,132]), degrade polycyclic aromatic hydrocarbons, and produce antibiotic substances [111,128]. Santoro and colleagues (2021) [9] have recently reported coral bleaching recovery and protection against mortality promoted by a probiotic consortium that included *Salinivibrio* sp., which is a member of the family *Vibrionaceae*. This *Salinivibrio* sp. strain was selected to comprise the consortium containing beneficial microorganisms for corals (BMCs) due to its antagonistic activity against *Vibrio coralliilyticus* and *V. alginolyticus*, both well-known coral pathogens [83,133]. The genomes from non-pathogenic and pathogenic members of the family *Vibrionaceae* can significantly differ [16], which explains the differential roles of specific genera within the same bacterial family in the coral holobiont.

Members of the family *Rhodobacteraceae* are also among the most ubiquitous bacterial groups in marine environments [134]. Some have been reported as indicators of thermal stress [135], while others have been described as opportunistic pathogenic bacteria that become more abundant when the coral host is under stress [49]. *Rhodobacteraceae* members are also known to establish mutualistic interactions with the coral host. An example of this is the coral-associated bacterium *Ruegeria*, which contributes to the settlement of larvae in octocorals, broadcast spawning and brooding in early stages of development [136], inhibition of pathogenic *Vibrio* strains [137], and degradation of toxic compounds [138]. The functional analysis of members of this family isolated from corals has also indicated their ability to degrade dimethylsulfoniopropionate (DMSP) [16], which has been previously described as a beneficial trait for corals [7,9,97].

Another abundant family was *Flavobacteriaceae*, found in 23 coral taxa. The family *Flavobacteriaceae* is widespread in the marine environment and is known to be associated with marine organisms [139,140,141]. Members of this family play an important role in the carbon cycle in the oceans; different studies have found numerous genes encoding for polysaccharide degradation [142,143], and more recently, a new species (*Flavihalobacter algicola*) capable of degrading alginate, the most abundant polysaccharide in brown algae, was identified [144]. Some *Flavobacteriaceae* members in corals, as well as other groups such as *Pseudomonadaceae* [145,146], are able to degrade DMSP [147], influencing sulfur compound metabolism and potentially generating sulfur-based antimicrobial compounds that inhibit the pathogenic effect of *Vibrio coralliilyticus* and *V. owensii* [148].

In addition, *Pseudomonadaceae*, *Endozoicomonadaceae*, and *Halomonadaceae* were reported in 27, 12 and 12 coral taxa, respectively (Figure 3b). Previous functional analyses revealed a high presence of genes that encode proteins involved in host-symbiont recognition and colonization in members of the *Pseudomonadaceae* and *Endozoicomonadaceae* families [16]. Other beneficial traits were also identified in that study, including antiviral defense in some *Endozoicomonas* strains, as well as the degradation of host-derived taurine, which were frequently found in strains of *Cobetia* and *Halomonas* (*Halomonadaceae*). The broad presence of these bacterial genera associated with different coral taxa suggests their potential importance in coral metabolism and health [36,149]. This is further corroborated by the probiotic thermal protective role of *Halomonas taeanensis* and *Cobetia marina*-related species strains (both belonging to the family *Halomonadaceae* [15]. Another study identified the genes involved in the degradation of DMSP in *Endozoicomonas* genomes and further confirmed that some strains can use DMSP as a carbon source, providing evindence that some members of this taxon play a central role in the sulfur cycle in corals [150] and can therefore be good BMC candidates [7,97], although this is not observed in all *Endozoicomonas* strains [36]. Other putative beneficial roles of *Endozoicomonas* include the protective action of *Symbiodinaceae* algae against bleaching pathogens [36,151]. *Alteromonadaceae*, which plays an important role in nitrogen provision to coral larvae in close relationship with Cyanobacteria [152,153], was also widely present across coral taxa (it was reported in 19 coral species).

Interestingly, some families were exclusively associated with a specific coral order. For instance, most of the exclusive bacterial families (*n* = 125) were associated with Scleractinian, whereas only 7 and 2 exclusive bacterial families were found for Alcyonacea and Capitata, respectively. Some studies have previously reported evidence of phylosymbiosis (i.e., microbiome diversity that correlates with host phylogenetics) [154] between corals and their associated microbiome (e.g., *Endozoicomonas* symbionts) in Scleractinian [23] and Gorgonians (Alcyonacea) [155]. In the Red Sea, the family *Weksellaceae* (Flavobacteriales) was the most frequently reported family among the exclusive families in Scleractinian corals, and therefore is a potential candidate to explore phylosymbiosis within these hosts. Members of this family have been associated with a vast range of habitats/organisms, including aquatic environments (e.g., seawater and marine sediments) and also birds, plants, and humans [156], although their association with corals has not been extensively documented. The most frequently identified family exclusively associated with Alcyonacea was *Coriobacteraceae* (Coriobacteriales). This family, which includes some benzene degraders [157], has been found in gut microbiota [158], the bacteriome of copepods [159], and even in association with whale sharks [160], although its role as part of the coral (*Alcyonaceae*) microbiome requires further investigation. For the order Capitata (hydrocorals), the two exclusive families *Neomegalonemataceae* (Rhodobacterales) and *Sphingosinicellaceae* (Sphingomonadales) were reported only in one study [80]. These families were recently described [161] and have been poorly explored.

Lastly, an important aspect to consider is that the total number of families per coral taxon was highly variable. As previously mentioned, different sequencing technologies were used in the reported studies used for the atlas, and this can contribute to an uneven sequencing depth among coral taxa and/or studies. For instance, NGS techniques can lead to finer data resolution in comparison to other technologies (e.g., Sanger) that were used in some of the studies [39,51,73]. However, some publications still reported a low associated bacterial diversity using NGS. This can be due to the use of different platforms (e.g., MiSeq vs. HiSeq), with a distinct final amount of data. The current reported bacterial diversity per coral taxa is highly variable and depends not only on the sequencing technology used and identification approach (cultivation-dependent vs. cultivation-independent techniques) but also on the sampling efforts and purpose of the study, which, in some cases, only targeted specific bacterial group(s), e.g., [36].

### 4.3. Bacterial Diversity among Red Sea Regions

We found differences in the reported bacterial taxa among the Red Sea regions considered for this survey, with the highest and lowest number of families found in the northern and southern regions, respectively. This finding may be related to the lower number of studies and sampling efforts being conducted in the southern Red Sea, and it could be biased by the different sequencing approaches used throughout the studies that were included for this survey, or even a combination of both.

Among the four bacterial families that have been exclusively reported in the southern Red Sea, *Acholeplasmataceae* is worthy of attention. Members of the *Acholeplasmataceae* family are commonly found in plants and animals and are thought to be pathogenic species due to their ubiquitous distribution and the fact that several of their strains have been isolated from diseased animals [162]. However, in 1987, Boyle and colleagues found the first association between a bacterium from this family and an aquatic invertebrate that was not related to a disease [163]. This family has been increasingly reported in the marine environment, and its abundance has been proven to be correlated with salinity [164,165,166].

*Nocardiopsaceae*, also exclusively reported in the southern Red Sea, encompasses thermophilic bacteria [167] and some halophilic or halotolerant groups, which have been isolated from hypersaline environments [168,169]. This family has also been described in the deep sea [170]. Some members of this family display antibacterial activity and the production of antibiotics and secondary metabolite compounds [171,172,173], suggesting that they are promising candidates for future surveys. *Thiotrichaceae* was also only present in the southern Red Sea, and several members of this family have been found to live in symbiotic relationships with invertebrates in hydrothermal vents [174]. However, their presence and roles in the coral microbiome need further research. The presence of these three families in the southern Red Sea and their apparent absence in both northern and central regions may be due to the harsher conditions in the South (e.g., higher temperatures), making it a more hostile and chemically challenging environment and hence more likely to present more resilient bacterial taxa. This could also potentially explain the previously mentioned lower bacterial diversity reported in this region when compared to the northern and central Red Sea, indicating that the bacterial diversity decreases as the environment becomes more extreme [95].

### 4.4. Cultured Bacterial Diversity

The cultured bacterial diversity found in this survey showed that most of the culturable bacteria were detected using cultivation-independent techniques, representing approximately 10% of the total families identified by this approach (33 bacterial families were identified using culture-dependent techniques vs. 324 that were detected using culture-independent techniques). Only two bacterial families (*Amorphaceae* and *Salinispharaceae*) were characterized using isolation approaches alone. The culturable coral-associated bacteria represent a small fraction of the entire bacterial community harbored in the coral host [16,50], which aligns with the still low percentage of culturable bacteria associated with diverse hosts and environments [175,176]. Culturing limitations are in part due to technical constrains in mimicking in situ conditions [177], despite recent major technological advances [178,179,180]. Hence, the cultured bacterial fraction present in Red Sea corals is still limited, and further efforts to optimize isolation methods that allow expansion of the culturable fraction are necessary to better uncover unculturable bacteria, the so-called “microbial dark matter” [181]. Understanding and exploiting bacterial isolate biochemical and biological traits contributes to further examination of prospective applications (e.g., ex situ manipulation of the coral microbiome, characterization and applications of bacterial secondary metabolites, etc.) [98]. Some of the bacterial groups reported here have potential beneficial roles in the holobiont, as previously mentioned, such as members of *Halomonas* (*Halomonadaceae*), *Pseudoalteromonas* (*Pseudoalteromonadaceae*), and *Cobetia* (*Halomonadaceae*) [15].

Interestingly, most of the coral taxa used for the isolation of their associated bacteria were soft corals, which might reflect their recognized potential as “biological factories” for a vast biotechnological repertoire [111]. However, hard corals have been more recently studied for the isolation of their microbial symbionts, as new conservation approaches emerge, where the use of coral probiotics is rapidly evolving as a nature-based innovative tool for coral therapies, according to their proposed beneficial mechanisms in the coral holobiont [7,96,97].

Further efforts to create a microbial isolate bank could contribute enormously to the isolate diversity repository from Red Sea corals. Currently, an initiative led by Peixoto and collaborators [182] is aiming to create the Red Sea Microbial Vault to isolate putative bacteria that could be used as coral probiotics for further research on coral restoration studies, among others. Such an initiative is aligned with the international Beneficial Microbes for Marine Organisms (BMMO) Network [183], which currently develops coral probiotic research in the Red Sea.

## 5. Conclusions

This review provides a comprehensive baseline survey of Red Sea coral-associated bacteria encompassing 20 years of research, showing an increased interest in this topic, especially during the past decade [103,184]. Our study is important to provide an overview of the bacterial diversity that lives in association with corals, while putting into perspective how little we actually know about certain locations and coral species in the Red Sea that are yet to be explored. Now, more than ever, it is crucial to dive into this topic, understanding the diversity and roles of symbiotic microbes, especially in naturally selected resistant holobionts, such as the corals from the Red Sea [185]. Microbial therapies are now proving to be extremely promising to improve coral tolerance to stress [9,11,15,17,99,101]. Thermally resilient corals adapted to extreme environmental temperatures are particularly promising candidates for further research to unveil their microbiome adaptations to high temperatures, among other stressors. Given the substantial loss of coral reef cover globally, the mitigation of local and global impacts on coral reefs will need to be combined with active restoration [186] and human-assisted/accelerated coral adaptation to global warming [98].

## Figures and Tables

**Figure 1 microorganisms-10-02340-f001:**
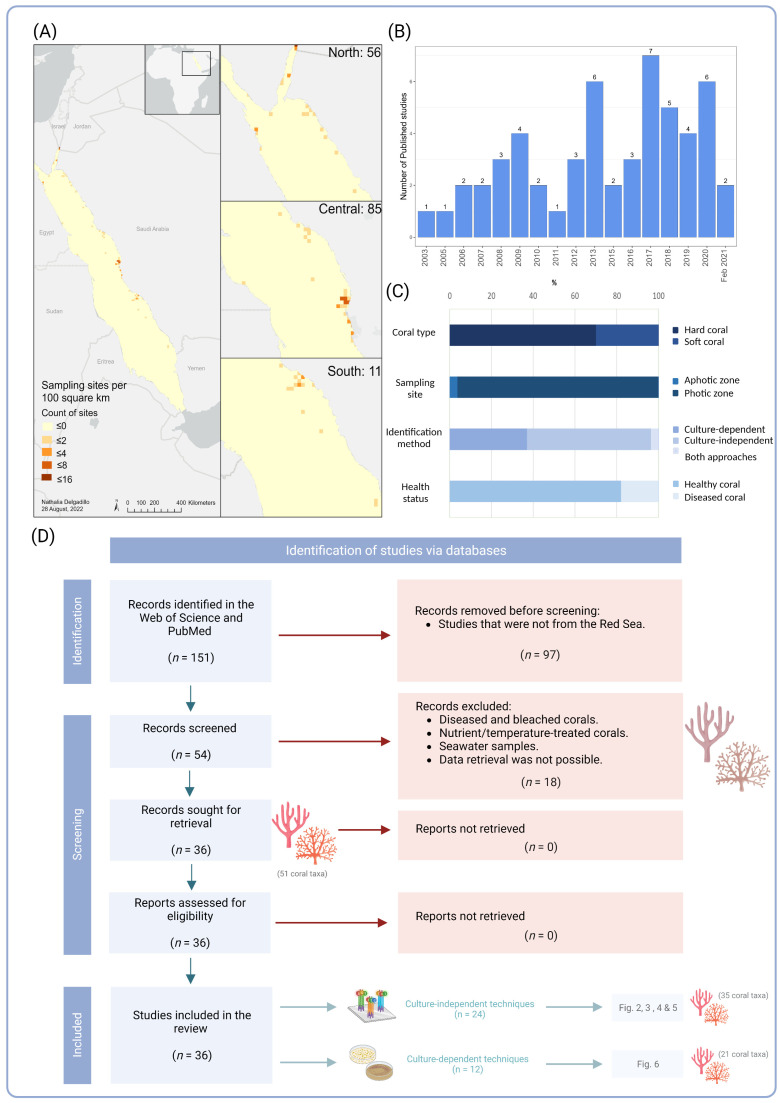
Systematic review of published studies investigating bacterial communities associated with corals along the Red Sea using a literature survey in the PubMed and Web of Science databases. (**A**) Map showing the spatial distribution and density of sampling sites of coral-associated bacteria along the latitudinal gradient of the Red Sea; (**B**) Bar graph indicating the number of published papers (*n* = 54) of studies in the Red Sea over the past two decades (2000—February 2021); (**C**) Bar graph representing coral groups (hard vs. soft), sampling depths (photic 0–50 m vs. aphotic zone 300–1000 m), study techniques (culture-dependent techniques vs. culture-independent techniques), and coral health state (healthy vs. unhealthy) used in the retrieved publications; (**D**) Flowchart following PRISMA guidelines, summarizing the retrieved publications, including the number of studied coral taxa and techniques used for the bacterial diversity Atlas.

**Figure 2 microorganisms-10-02340-f002:**
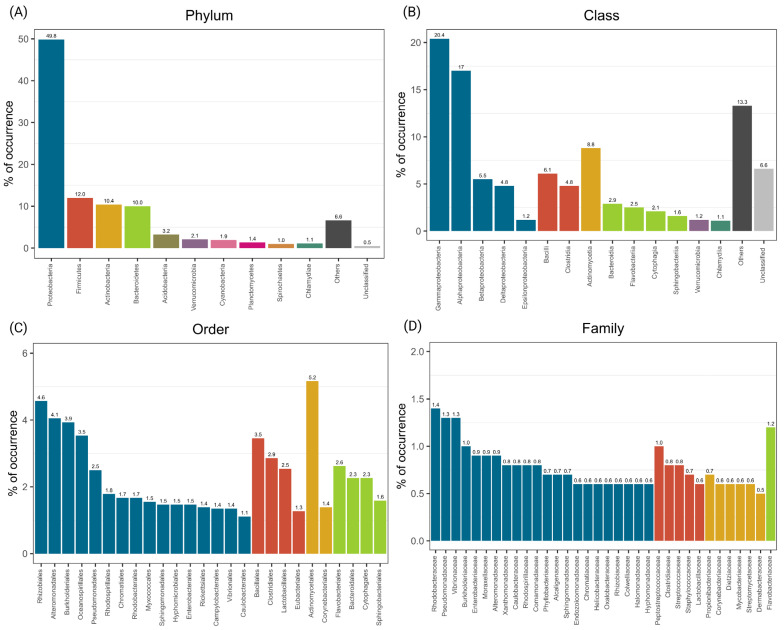
Coral-associated bacteria across all coral species and studies, grouped at four different taxonomic levels. (**A**) Phylum; (**B**) Class; (**C**) Order and (**D**) Family. For each graph, the Y-axis represents the percentage of occurrence. Groups with the lowest occurrence in the different taxonomic levels were grouped into “Others”. Unclassified groups and “Others” for order and family levels are available in Appendix A. Colors used for class-, order- and family-level graphs represent members of the same phylum.

**Figure 3 microorganisms-10-02340-f003:**
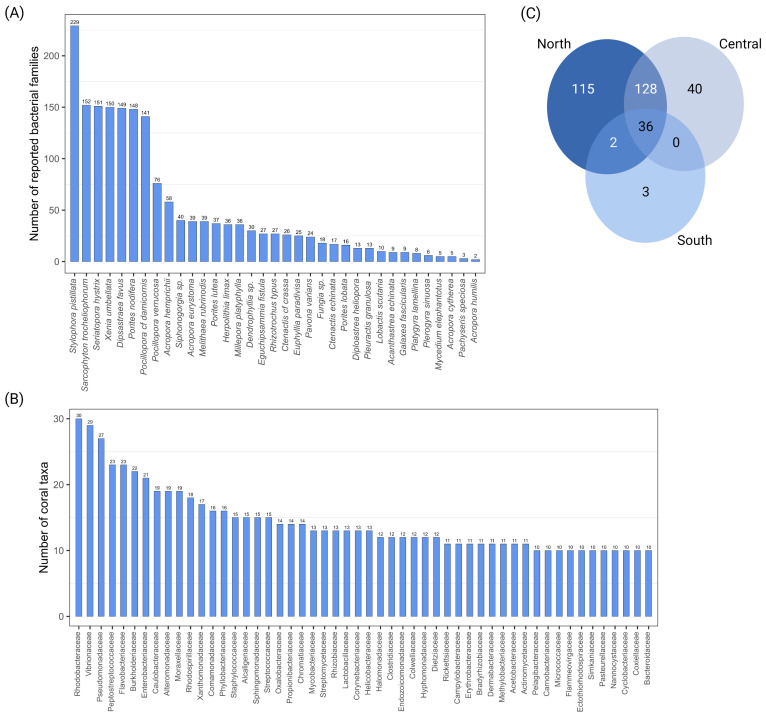
(**A**) Bacterial diversity (by number) at the family level for 35 coral taxa according to data retrieved from the selected publications in this study. Seven coral species showed a higher number of bacterial families (*n* > 130). (**B**) Bacterial families reported in 10 or more coral taxa. (**C**) Venn diagram showing the number of common and exclusively reported bacterial families among Red Sea regions.

**Figure 4 microorganisms-10-02340-f004:**
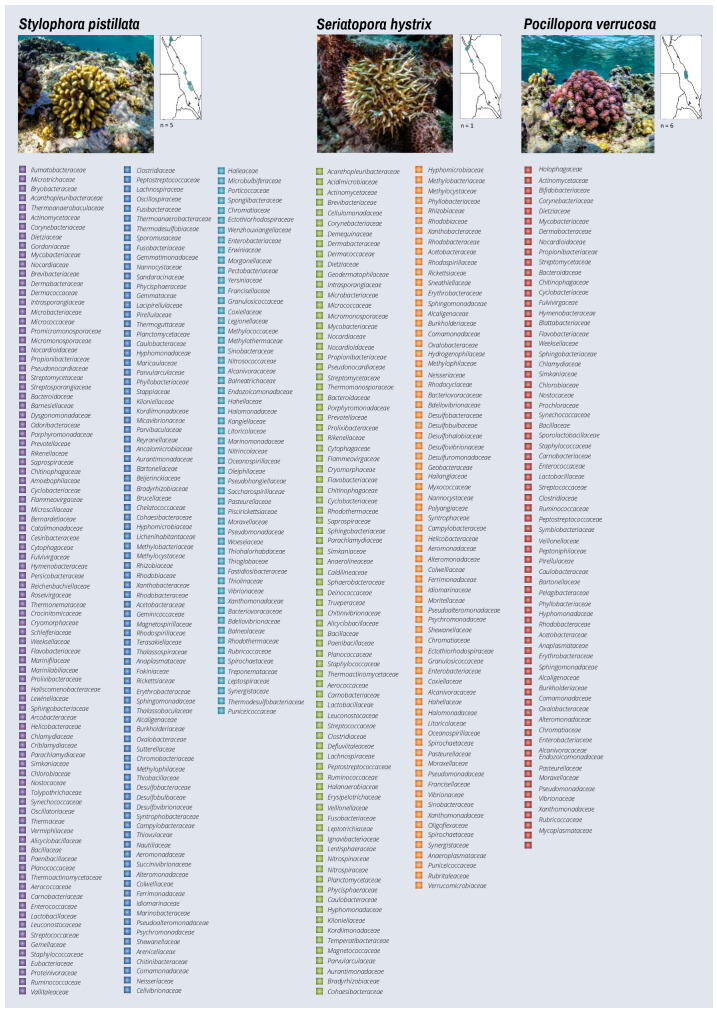
The Red Sea coral-associated bacterial Atlas: Coral-associated bacteria across Red Sea coral species. The reported families are shown. Due to space constraints, coral hosts are not presented by taxonomic classification, but rather by the overall abundance of associated bacterial families. The number of studies included for the Atlas, per coral species (*n*), and the corresponding location of the study sites in the Red Sea are indicated in the map inset close to the image in vivo.

**Figure 5 microorganisms-10-02340-f005:**
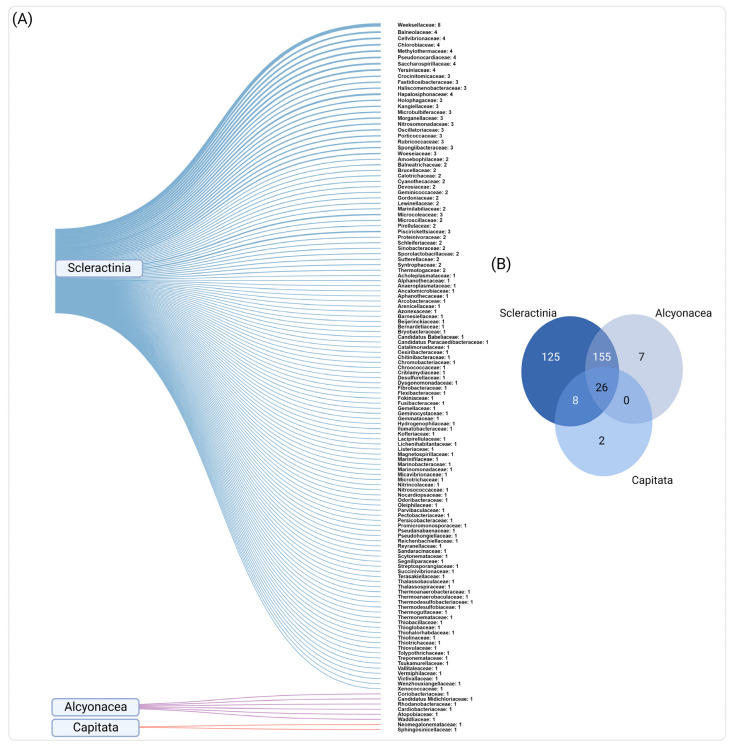
Bacterial families exclusively identified according to the coral order. (**A**) Sankey diagram showing the families exclusively associated with hard corals and hydrocorals (*Scleractinia* and *Capitata*) and soft corals (*Alcyonacea*). The number each bacterial family was reported across studies is shown. (**B**) Venn diagram showing the number of common and exclusively reported bacterial families among coral orders.

**Figure 6 microorganisms-10-02340-f006:**
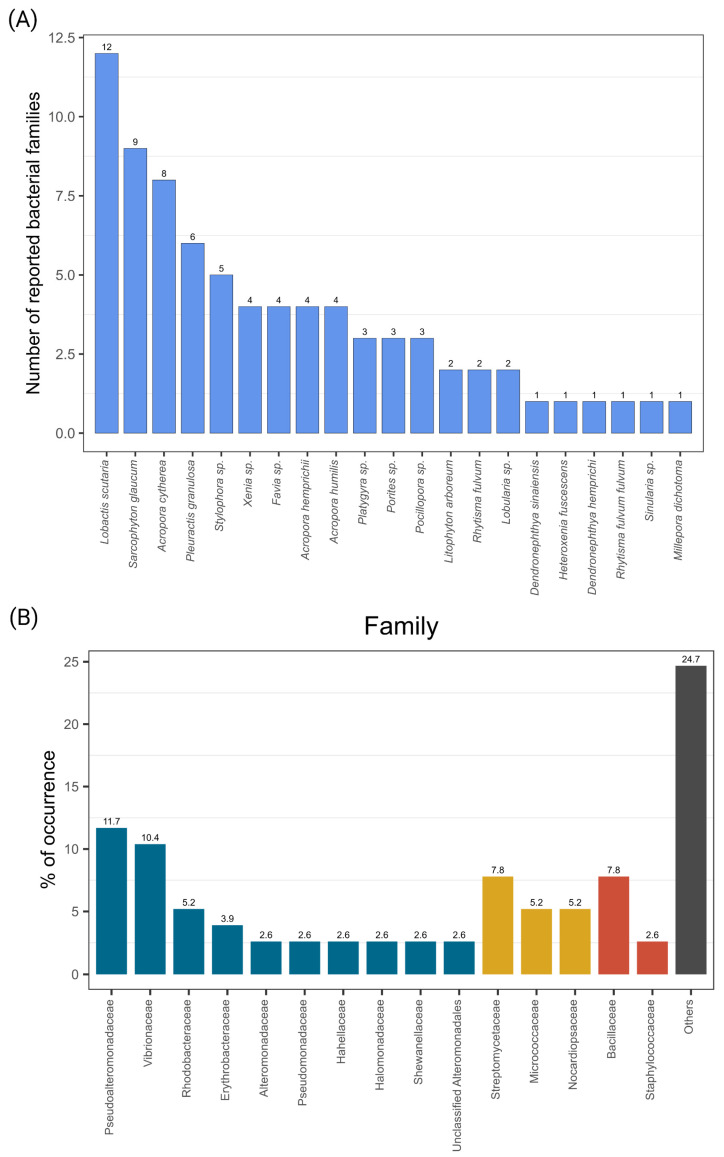
Coral-associated bacteria based on the examined culture-dependent studies. (**A**) Number of reported bacterial families associated with different coral species. (**B**) Frequency of bacterial diversity at the family level. Bacterial families are color-coded per phylum following Figure 2. Families with a lower frequency were grouped under “Others”.

**Table 1 microorganisms-10-02340-t001:** Retrieved studies from bacterial communities associated with Red Sea corals from the PubMed and Web of Science databases. The table includes year of publication, reference, collection region (N: North; C: Central; S: South), host taxa, accession numbers, data deposition, and microbial identification approach (CDT: Culture-dependent technique; CIT: Culture-independent-technique).

Healthy Corals						
Year	Reference	Red Sea Region	Host	Accession Number	Data Deposition	Microbial Identification Approach
2006	[50]	N	*Lobactis scutaria*	DQ107384-DQ107405	NCBI	CDT
2007	[39]	N	*Pleuractis granulosa*	DQ117312-DQ117435	NCBI	CIT
2008	[51]	N	*Fungia* sp.	AM941168—AM941189	NCBI	CIT
2008	[52]	N	*Lobactis scutaria* and *Platygyra lamellina*	EF466016–EF466058; EF576992–EF577027	NCBI	CIT
2008	[53]	N	*Pleuractis granulosa*	DQ097300	NCBI	CDT
2009	[54]	N	*Platygyra* sp., *Porites* sp., *Pleuractis granulosa*, *Favia* sp., *Stylophora* sp., *Pocillopora* sp., *Rhytisma fulvum fulvum* and *Xenia* sp.	FJ041063—FJ041108	NCBI	CDT
2009	[55]	N	*Pleuractis granulosa*	DSM 45190	NCBI	CDT
2010	[56]	N	*Sinularia polydactyla*	-	-	CDT
2010	[57]	N	*Favia* sp.	FJ041083	NCBI	CDT
2011	[58]	N	*Acropora eurystoma*	GU319121–GU319777	NCBI	CDT/CIT
2012	[59]	C	*Pocillopora verrucosa*, *Astreopora myriophthalma*, *Stylophora pistillata*, *Sarcophyton* sp. 1 and *S.* sp. 2	SRA012656	NCBI	CIT
2012	[60]	N	*Platygyra* sp., *Porites* sp., *Stylophora* sp., and *Pocillopora* sp.	FJ041071; FJ041068; FJ041064; FJ041069; FJ041065	NCBI	CDT
2012	[61]	N	*Dendronephthya sinaiensis*, *Dendronephthya hemprichi*, *Xenia* sp., *Lobularia* sp., and *Rhytisma fulvum*	JF292923—JF292930	NCBI	CDT
2013	[49]	N and S	*Ctenactis cf crassa* and *Herpolitha limax*	SAMN01939168–SAMN01939191	NCBI	CIT
2013	[62]	N	*Millepora dichotoma*	HQ288552- HQ288737	NCBI	CDT
2013	[63]	C	*Acropora hemprichii*	SRA062645	NCBI	CIT
2013	[64]	S	*Stylophora pistillata*, *Acropora humilis* and *Acropora damicornis*	PRJNA189184 and KC668414 to KC669277	NCBI	CIT
2013	[65]	N	*Sarcophyton glaucum*	JQ929053-JQ929072	NCBI	CDT
2015	[66]	C	*Ctenactis echinata*	PRJNA277291	NCBI	CIT
2016	[67]	C	*Pleuractis granulosa*	PRJNA282461	NCBI	CIT
2016	[68]	C	*Acropora hemprichii* and *Pocillopora verrucosa*	PRJNA287432	NCBI	CIT
2017	[36]	C	*Stylophora pistillata*, *Pocillopora verrucosa*, and *Acropora humilis*	PRJNA323666	NCBI	CIT
2017	[37]	C	*Stylophora pistillata* and *Pocillopora verrucosa*	PRJNA280923	NCBI	CIT
2017	[69]	C	*Porites lobata*	PRJNA352338	NCBI	CIT
2017	[70]	C	*Eguchipsammia fistula*	PRJNA354830	NCBI	CIT
2017	[71]	N and C	*Dendrophyllia* sp., *Eguchipsammia fistula*, and *Rhizotrochus typus*	PRJNA354830	NCBI	CIT
2017	[72]	C	*Pocillopora verrucosa*	PRJNA335276	NCBI	CIT
2018	[73]	N	2 soft corals and 4 hard corals (coral ID not provided in the paper)	MH040868, MH040869, MH040870, MH040871, MH045063, MH045064, MH045263, MH045264, MH046783 and MH046784	NCBI	CDT
2018	[25]	C	*Pocillopora verrucosa*	PRJNA394597	NCBI	CIT
2018	[74]	N	*Heteroxenia fuscescens*, *Litophyton arboerum*, *Sarcophyton trocheliophorum*, *Sinularia* sp., *Sacrophyton acutum* and *Lobophytum pauciflorum*	MG757677.1 MG757675.1 MG757672.1 MG757678.1	NCBI	CDT
2019	[75]	N	*Rhytisma fulvum fulvum*	-	-	CDT
2019	[18]	C	*Acropora hemprichii* and *Pocillopora verrucosa*	PRJNA491299	NCBI	CIT
2019	[76]	N	*Stylophora pistillata*	-	-	CIT
2020	[21]	N	*Porites nodifera*, *Dipsastraea favus*, *Pocillopora cf damicornis*, *Seriatopora hystrix*, *Xenia umbellata*, and *Sarcophyton trocheliophorum*.	PRJNA509355	NCBI	CIT
2020	[77]	C	*Melithaea rubrinodis* and *Siphonogorgia* sp.	SRP115467	NCBI	CIT
2020	[78]	N	*Sarcophyton glaucum*	-	-	CDT
2020	[79]	N	*Euphyllia paradivisa*	SRP133996	NCBI	CIT
2020	[80]	C	*Acanthastrea echinata*, *Diploastrea heliopora*, *Fungia* sp., *Galaxea fascicularis*, *Mycedium elephantotus*, *Pavona varians*, *Plerogyra sinuosa*, *Pocillopora verrucosa*, *Porites lutea*, *Stylophora pistillata*, *Millepora platyphylla*, *Acropora cytherea*, *Pachyseris speciosa* and *Xenia* sp.	PRJNA437202	NCBI	CIT
2021	[81]	N and C	*Stylophora pistillata*	PRJNA681108	NCBI	CIT
2021	[82]	N	*Styllophora pistillata*	PRJNA674053	NCBI	CIT
2021	[16]	N and C	*Acropora hemprichii*, *Acropora humilis*, *Acropora cytherea* (*Cardénas* and *Voolstra*, 2021)	GS0145871; PRJNA343499	NCBI	CDT/CIT
Diseased Corals						
2003	[83]	N	*Pocillopora cf damicornis*	-	-	CDT
2005	[84]	N	*Goniastrea* sp. and *Dipsastraea favus*	AY643537	EMBL	CDT
2006	[85]	N	*Dipsastraea favus*	CBMAI 722	NCBI	CDT
2007	[86]	N	*Favites* sp. and *Favia* sp.	EF089403–EF089533, EF433087–EF433174, EF089519; EF089534–EF089544	NCBI	CIT
2009	[87]	N	*Favia* sp.	FJ210722	NCBI	CDT
2009	[88]	N	*Favia* sp.	GQ215061–GQ215096; GQ215097–GQ215227	NCBI	CDT
2013	[89]	N	*Millepora dichotoma*	HQ288781	NCBI	CDT
2015	[90]	N	*Favia* sp.	4541470.3–4541481.3.	MG-RAST	CIT
2018	[91]	C	*Astreopora* sp., *Coelastrea* sp., *Dipsastraea* sp., *Goniopora* sp., *Montipora* sp., *Pavona* sp., *Platygyra* sp., *Psammocora* sp., and *Gardineroseris* sp.	PRJNA436216	NCBI	CIT
Seawater and Microbial Biofilms Associated with Red Sea Coral Reefs						
2016	[92]	C	Seawater and microbiome biofilm in Terracotta tiles	PRJNA306204	NCBI	CIT
2017	[93]	C	Seawater from coral reefs	PRJNA357506	NCBI	CIT
2018	[94]	C	Seawater from coral reefs	PRJNA352340	NCBI	CIT
2019	[95]	N, C and S	Seawater and microbiome biofilm in Autonomous Reef Monitoring Structures (ARMS)	PRJNA479721	NCBI	CIT

## Data Availability

The datasets analyzed for this study can be found in the NCBI database (https://www.ncbi.nlm.nih.gov/ (accessed on 1 February 2021)) and in the original publications. Please refer to Table 1 for specific accession numbers of each publication.

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
