# Peer review of "Red Sea Atlas of Coral-Associated Bacteria Highlights Common Microbiome Members and Their Distribution across Environmental Gradients—A Systematic Review"

_microorganisms, 2022, doi:10.3390/microorganisms10122340_

Round 1

Reviewer 1 Report

The current results are dependent on Taxonomy Analysis too in NCBI, and the authors did not reanalyze the data. It seems not proper to represent data in this kind of method. There are still lots of sequences are assigned to unidentified. Therefore, the results can not fully reflect the coral holobiont in Red Sea.

Author Response

Dear Editor,

We would like to thank you for the opportunity to revise our manuscript “Red Sea Atlas of coral-associated bacteria highlights common microbiome members and their distribution across environmental gradients” to be considered for publication at Microorganisms. Also, we would like to thank the anonymous reviewers for the insightful comments and suggestions that were helpful to improve the manuscript.

In the below section, we provide a point-by-point response where we address all issues and suggestions raised by the reviewers. The revision is approved by all authors and changes are tracked in the attached manuscript.

Please do not hesitate to get in touch if you have any questions or concerns.

Reviewer 1: The current results are dependent on Taxonomy Analysis tool in NCBI, and the authors did not reanalyze the data. It seems not proper to represent data in this kind of method. There are still lots of sequences are assigned to unidentified. Therefore, the results cannot fully reflect the coral holobiont in Red Sea.

Response: Thank you for this comment, we have highlighted a few points in our manuscript that may clarify the rationale for us not to reanalyze the data, as detailed below:

  • This is a review paper, not a research article. As the available information to date is scattered in the literature, our intent is to provide an updated reference tool from which patterns and gaps can be more clearly evidenced and future studies can draw from. Our goal was to report bacterial groups from microbiome studies that have been described in the Red Sea, according to the literature, in a systematic way. This would fill a gap in knowledge on the bacterial taxa that exist across coral reefs in this region while minimizing potentially biased conclusions that could be reported if we had decided to reanalyze the data, as discussed below.

  • We used the Taxonomy Analysis tool in NCBI, which provides curated classification and nomenclature for all of the organisms in the public sequence databases. Also, NGS sequences are deposited as raw sequence data in the NCBI database, not associated with taxonomy. NCBI aligns NGS reads against a pre-computed k-mer dictionary databases to provide bacterial taxonomy to sequences. Therefore, our data does provide taxonomym independent from those provided by the authors, that may vary between versions of SILVA/Greengenes databases. This highlights that even reanalysis of the collected microbiome data would contain unclassified bacteria, because they have not been yet to be identified in any database. Please see more details here:

https://www.ncbi.nlm.nih.gov/sra/docs/sra-taxonomy-analysis-tool/

  • Although we agree that the reanalysis of the microbiome data potentially may provide novel insights, this approach also has several constraints, which include; i) limits our review to few papers that used Illumina sequencing technique, ii) excludes 454-pyrosequencing due to bioinformatics limitations to compare between two different sequencing technologies, and iii) neglects the culture-depended studies, which would also exclude a technique that may be popular in low-income countries surrounded the Red Sea. Together, this would bias our results toward a specific technology and ignore other important studies that have been done in the region.

  • Comparing/reanalyzing microbiome data from different studies that have been subjected to different library preparation, sequencing depth and protocols, sampling procedures, and many other variables may lead to even more biased results. This would produce unreliable alpha and beta diversity, which requires extensive statistical and bioinformatics validation. Most studies confirmed that reanalyzing microbiome data from different studies is inconsistent, even from within the same lab studies. Please see the following studies.

https://www.frontiersin.org/articles/10.3389/fmicb.2021.740932/full

https://www.ncbi.nlm.nih.gov/pmc/articles/PMC4523815/

https://www.ncbi.nlm.nih.gov/pmc/articles/PMC3398667/

https://bmcmicrobiol.biomedcentral.com/articles/10.1186/s12866-017-1101-8

  • It is worthwhile to note here that we used the criteria of presence/absence of bacteria in our analysis, and we did not use abundance data, which is highlighted in the text. We believe that this is the best approach to report what has been described in other studies, and avoid biased conclusions and new assignments. Because the goal of the study is to provide a catalog of microbiome diversity (i.e., an Atlas), it needs to be inclusive. These goals and limitations are highlighted in the new version of the manuscript.

Reviewer 2 Report

 This review provides a comprehensive baseline survey of the Red Sea

coral-associated bacteria encompassing 20 years of researches. The Red Sea  is a suitable model for studying coral reefs under climate change due 

to its strong environmental gradient that provides a window into future global warming scenario. therefore, the baseline features of microbiome associated with the Rea Sea coral is important for understanding the adaptation of coral in the light of bacteria under future climate changes.  This review is well designiated and well wrote, thus I thought this review article is valuable for publication in Microoganisms.  However, some minor issues still need to be improved. 

1.  line 97: one of the words "conducted" shoud be deleted. 

2. line 99: " using the Web of Science and Pub- 98 Med public databases from July 2020 until February 2021". In this sentence,  the literature retrieval time need check (I think it must be wrong), as in Table 1, the paper published in 2003 is included and this review, as you say, represent publised paper of over the past two decades.  

Author Response

Dear Editor,

We would like to thank you for the opportunity to revise our manuscript “Red Sea Atlas of coral-associated bacteria highlights common microbiome members and their distribution across environmental gradients” to be considered for publication at Microorganisms. Also, we would like to thank the anonymous reviewers for the insightful comments and suggestions that were helpful to improve the manuscript.

In the below section, we provide a point-by-point response where we address all issues and suggestions raised by the reviewers. The revision is approved by all authors and changes are tracked in the attached manuscript.

Please do not hesitate to get in touch if you have any questions or concerns.

Reviewer #2

Reviewer: This review provides a comprehensive baseline survey of the Red Sea coral-associated bacteria encompassing 20 years of research. The Red Sea is a suitable model for studying coral reefs under climate change due to its strong environmental gradient that provides a window into future global warming scenario. therefore, the baseline features of microbiome associated with the Rea Sea coral is important for understanding the adaptation of coral in the light of bacteria under future climate changes. This review is well designiated and well wrote, thus I thought this review article is valuable for publication in Microoganisms. However, some minor issues still need to be improved.

Response: We would like to thank the reviewer for the constructive feedback.

Reviewer: line 97: one of the words "conducted" should be deleted.

Response: The repeated word was deleted.

Reviewer: line 99: " using the Web of Science and Pub- 98 Med public databases from July 2020 until February 2021". In this sentence, the literature retrieval time need check (I think it must be wrong), as in Table 1, the paper published in 2003 is included and this review, as you say, represent published paper of over the past two decades.

Response: Thank you for pointing this out. This was a mistyping and it should be from 2002 not 2020. We corrected it.

Reviewer 3 Report

Abstract

L. 16: In “Coral holobionts in the Red Sea live thrive under high water temperatures in the summer, but also under strong seasonal fluctuations, high salinity (41 PSU), and in oligotrophic seawater.” remove “live”, please. I am sure it is not the only place that present high water temperature in the summer. Strong seasonal fluctuations such as?

L. 20: why only bacteria as the widespread NGS primer-pair system used assess bacteria and archaea?

L. 23-24: Why these species were the most studied? Easy access? More abundant?

Introduction

L. 35-37: Why it was quantified?

L. 53-55: Besides, vertical transmission, why corals can acquire they food if they are not filter-feeders? Through food? If so, how can the host “selects” what is nutrients from what can assist the host in its needs, such as in physiology?

L. 63-65: “long-term mean summer temperature” comprehend a few years, or one decade or two, as the review focuses on the 20 years of research into the Red Sed? Please, clarify. Provide the salinity from the south region, please? So one can be certain that the higher salinity in the north is indeed higher.

L. 65-67: In “Similarly, the chlorophyll concentration and attenuation coefficients are higher in the south-66 ern Red Sea due to exchange with nutrient-rich seawater from the Gulf of Aden [30].” Chlorophyll concentration I understand that it is related to photosynthesis, but attenuation coefficients I have no idea what it measures and why it is important. Please, explain.

L. 69: In “high gene flow and genetic similarity” can it be related to reproduction? How corals reproduce? Only sexually?

L. 80-83: Why do the authors think that the Red Sea studies still lower that other regions? Permission to collect the samples could be one reason?

L. 89-90. I believe the authors should state since the beginning of the introduction that the focus of this review would be bacteria, instead of prokaryotes or microbiome (L. 38). I was curious to see if there was any study that managed to isolate archaea from corals and assess their functional capacity through genome sequencing. Note that, the term microbiome could still be using, it just needs to made it clear that it is related to bacteria and that archaea are not included in the study. It would also be very interest if archaea were also investigated, because usually, studies involving bacteria surpass the ones investigating any other group of microorganisms. 

MM

L. 101-105. How can one be sure that using only five words in the search engineer would be enough to comprehensively access all the relevant literature??? There is a R package named litsearchr (Grames et al., 2019) that could assist in a more broadly search. I suggest the authors give it a try.

L. 111-115: Were the authors able to collect all these information from the relevant literature? Whether any of the relevant meta-data was not available, would the article still be included in the meta-analyses? Additionally, sequencing technology would not be just culture-based vs sequencing, but also 454-pyrosequencing vs illumine, correct?

L. 115-119: How? Name of the sampling site? Please, clarify.

L. 125-126: I know that corals are not filter feeders, but seawater microbiome would not serve as a way to determine what was specific in coral species?

L. 128-131: Why the culture-independent and -dependent were considered separately?

L. 137-138: I would strongly suggest to re-analyze the whole dataset. Considering that the results were searched from the last 20 years and bacterial taxonomy changed over the years, it does not make sense to compile an atlas which have so many distinct classifications. 

L. 139-143: I believe that articles with Sanger sequencing were involved in the culture-dependent or perhaps clone library or DGGE band, correct? How can it be compiled together with NGS, when fragments are from distinct size and 16S rRNA gene regions? Above it was stated that two datasets were analyzed (culture-independent and -dependent), how sanger and NGS were combined for the atlas? Even for pipelines over time, there were distinct versions of Silva, for instance, including changes in the bacteria classification along the two decades span. There is also a sharp contrast between OTU and ASV, how can it be ignored. I can not see the meaning of doing such a work, when so many different inputs were combined just to have a list of bacteria associated with corals. What can it add to the literature available? The authors have already all the data. Also, they have the expertise to do a meaningful analysis of the data and provide a comprehensive review of the bacteria associated with corals. Besides, why remove the duplicate, when it can provide how abundant a taxon was in a coral species and consequently their putative function?

Author Response

Reviewer #3

Abstract

Reviewer: L. 16: In “Coral holobionts in the Red Sea live thrive under highw temperatures in the summer, but also under strong seasonal fluctuations, high salinity (41 PSU), and in oligotrophic seawater.” remove “live”, please. I am sure it is not the only place that present high-water temperature in the summer. Strong seasonal fluctuations such as?

Response: Thanks for pointing this out. We removed “live”, it was a typo indeed.

This sentence was edited to avoid confusion and clarify the context. Corals in the Red Sea have evolved and live in environmental conditions which have been defined extreme, especially due to the high water temperature and salinity. Moreover, strong seasonal fluctuations of several environmental variables contribute to the overall harsher conditions that the coral holobiont faces in the Red Sea compared to other bodies of water, on top of the contrasted thermal and environmental regimes along the latitudinal gradient of the Red Sea. This is also explained in light of the relevant literature in the introduction section (Line 61-80).

https://journals.plos.org/plosone/article?id=10.1371/journal.pone.0246854

“Coral holobionts in the Red Sea have adapted to an overall unique marine environment characterized by a combination of conditions deemed extreme both in shallow and deep water compared to other regions. In the shallow photic zone, Red Sea corals seasonally face salinity of 41 PSU and temperatures of 32oC. In the aphotic zone, the water temperature never drops below 21oC and salinity remains around 40.5 PSU.”

Reviewer: L. 20: why only bacteria as the widespread NGS primer-pair system used assess bacteria and archaea?

Response: Although we agree archaea are also interesting members of the coral microbiome, most of the retrieved data from Red Sea studies focused only on bacteria (and most of the primers used don’t amplify archaea so well, but are used to avoid the amplification of chloroplast 16S). Consequently, for this systematic review of 20 years of literature combined into the Atlas, we decided to limit our analyses to the most extensively studied bacterial communities.

Reviewer: L. 23-24: Why these species were the most studied? Easy access? More abundant?

Introduction

Response: We report the result of our literature survey here, and highlight the low number of coral species studies along the Red Sea also to emphasize the gap in knowledge. It is hard to define the reasons why these species were the most studied. They may range from abundance, availability, and ability of each specific team to identify and access these species. We edited the sentence in the abstract, to highlight that this is just a portion of the diversity:

“Our data also showed that, despite the high diversity of corals in the Red Sea, the most studied corals were Pocillopora verrucosa, Dipsastraea spp., Pleuractis granulosa, and Stylophora pistillata.”

and also added a sentence to speculate the reasons:

“The most commonly studied corals (e.g, Pocillopora verrucosa, Dipsastraea sp., Pleuractis granulosa, and Stylophora pistillata) are hard corals (subclass: Scleractinia) and are widely distributed in the Red Sea [47]. The reasons for the selection of these species are difficult to define, it may vary from a random selection to higher abundance in specific sampling sites or the ability of each specific team to identify and access these species, among other potential explanations. Besides, Scleractinian corals are commonly found in mid-shallow waters (i.e., 0-30 m depth), making them good candidates for coral microbiome studies, as they are relatively easy to spot and manipulate in ex situ conditions [57,58,60,64]. Additionally, in the Red Sea, scleractinian coral diversity is higher than soft coral diversity in mid-shallow waters [65,66].”

Reviewer: L. 35-37: Why it was quantified?

Response: We want to highlight the importance of photosynthetic endosymbionts associated with corals and how they are the main source of energy (by means of photosynthates) for most of Scleractinian corals (up to 90%). This introduces the reader to the microorganisms and how they play a vital role for corals’ survival.

Reviewer: L. 53-55: Besides, vertical transmission, why corals can acquire they food if they are not filter-feeders? Through food? If so, how can the host “selects” what is nutrients from what can assist the host in its needs, such as in physiology?

Response: Although most shallow corals obtain the majority of food from the photosynthetic endosymbiont, corals can also obtain food heterotrophically (to different extents), via filter feeding or capturing plankton using their tentacles. Selection of specific microbes may be attributed to several reasons: i) transmission mode, as the reviewer suggested, ii) environmental selection for microbes that are a better fit for the environmental regime and succession over other bacterial populations, iii) biochemical composition of mucus, which provides essential nutrient for certain types of bacteria and then act as a selective barrier, iv) recognition mechanism (e.g., chemotaxes) between the host and associated bacteria, or v) tissue chemistry and immune status that may promote growth for specific bacterial phylotypes. However, microbiome selection remains an active topic of research. Some studies have indicated different levels of microbiome flexibility as a response to environmental changes over short time frame and according to different coral species. This suggests that the selection of specific microbes that are more suitable for specific environmental conditions may be a key adaptive mechanism for corals/animals to survive under different environmental regimes, although the actual mechanism for its selection has yet to be fully explored. We allude to this in the following sentence:

“Therefore, understanding the composition, diversity, and dynamics of microbiomes associated with corals across environmental settings is key to understanding the potential role of the microbiome in improving coral acclimation to environmental stressors, including those derived from ongoing ocean warming.”

Reviewer: L. 63-65: “long-term mean summer temperature” comprehend a few years, or one decade or two, as the review focuses on the 20years of research into the Red Sed? Please, clarify. Provide the salinity from the south region, please? So one can be certain that the higher salinity in the north is indeed higher.

Response: This is a good question. In fact, there is no standard time frame for the “long-term mean summer temperature”, which is calculated using remote sensing data. It depends on the time frame of the available remote sensing dataset(s). The SST average reported in our manuscript is based on remote sensing data for 30 years (Please see Osman et al. 2018).

Salinity average for the southern Red Sea was added in the MS, as requested:

“The seawater temperature increases southward (long-term mean summer temperature ranges from 25 to 32°C) [28], while salinity is higher in the north (41 PSU) than in the south (36 PSU), due to a higher evaporation rate and lack of freshwater input [29].”

Reviewer: L. 65-67: In “Similarly, the chlorophyll concentration and attenuation coefficients are higher in the south-66 ern Red Sea due to exchange with nutrient-rich seawater from the Gulf of Aden [30].” Chlorophyll concentration I understand that it is related to photosynthesis, but attenuation coefficients I have no idea what it measures and why it is important. Please, explain.

Response: Attenuation coefficient is a proxy to measure light penetration (i.e., visibility or water transparency). It is a remote sensing product that can assess water visibility over large scale. Here, we wanted to inform the reader that the southern Red Sea has higher chlorophyll concentration and low seawater visibility which means less light compared to central and northern Red Sea. To avoid confusion, we edited the sentence and removed “attenuation coefficient” to make it clear:

“Similarly, the chlorophyll concentration is higher in the southern Red Sea due to exchange with nutrient-rich seawater from the Gulf of Aden, and consequently, seawater visibility in the south is lower than in the central and northern Red Sea regions [30].”

Reviewer: L. 69: In “high gene flow and genetic similarity” can it be related to reproduction? How corals reproduce? Only sexually?

Response: Yes, this is related to reproduction. Corals reproduce both sexually and asexually. Through budding, corals can build their colony carbonate skeleton, while sexual reproduction is a way to exchange genetic material with other colonies and occupy new reef areas. We edited the sentence to improve the clarity.

“Despite the contrasting environmental conditions, genetic analysis of coral species to date have shown high gene flow and genetic similarity between coral populations along the latitudinal gradient of the Red Sea, suggesting that coral hosts have low genetic variations along the Red Sea latitudinal gradient [31,32].”

Reviewer: L. 80-83: Why do the authors think that the Red Sea studies still lower that other regions? Permission to collect the samples could be one reason?

Response: The low number of publications is an observed pattern along the Red Sea (see Berumen et al., 2013 – Reference #62). The Red Sea is surrounded by eight countries, most of them are low-income countries. Until recently, most studies were published by other countries, although Saudi Arabia dominated the publications during the last decade. Research from other countries remain limited, which is, in fact, part of our results as well (please see Figure 1) and the paragraph below:

“Another interesting aspect is that the studies are mostly aggregated in certain parts of the Red Sea, the majority of sampling efforts have been carried out in the north and central regions along the Saudi Arabian Red Sea coast, leaving a vast territory unexplored. In addition, the studies in the northern Red Sea have mostly been performed in the Gulf of Aqaba. In contrast, the southern and western regions show a gap in microbiome research, with only a few publications available (see Table 1). This lack of information was reflected in the reported bacterial diversity throughout the Red Sea: only 41 bacterial families were reported to be associated with corals in the south, unlike the northern and central regions with a remarkably higher number of reported families (n=282 and 204, respectively). This highlights the urgent need to expand microbiome research in unexplored areas along the Red Sea coastline.”

Reviewer: L. 89-90. I believe the authors should state since the beginning of the introduction that the focus of this review would be bacteria, instead of prokaryotes or microbiome (L. 38). I was curious to see if there was any study that managed to isolate archaea from corals and assess their functional capacity through genome sequencing. Note that, the term microbiome could still be using, it just needs to made it clear that it is related to bacteria and that archaea are not included in the study. It would also be very interest if archaea were also investigated, because usually, studies involving bacteria surpass the ones investigating any other group of microorganisms.

MM

Response: This is a really interesting point and, again, we agree that archaea are an important component of the microbiome. However, none of the retrieved studies, to our knowledge, focused on archaea along the Red Sea (and most of the primers used don’t amplify archaea well, but are used to avoid the amplification of the chloroplast 16S). Thus, it does not fit the aim of this work, which was to comprise the 20 years of what has been described in the Red Sea. We did clarify in the new version of the text that we wanted to assess and summarize the information reported in the literature.

Reviewer: L. 101-105. How can one be sure that using only five words in the search engineer would be enough to comprehensively access all the relevant literature??? There is a R package named litsearchr (Grames et al., 2019) that could assist in a more broadly search. I suggest the authors give it a try.

Response: Thank you for your suggestion. Although “litsearchr” package on R is an interesting tool for meta-analysis, this package cannot replace or supplement the search of the web of science. In fact, we must search for literature on a given database first (e.g., web of science, Scopus, etc), and export the search results that can be then used by the package to look and/or quantify specific terms within the imported database. “Web of Science” is a standardized tool to quantify and search for research that is used by most researchers. According to our previous experience and trials in other databases (e.g., Scopus, BASE or other libraries), Web of Science was the best database in retrieving the highest number of publications using our search words, also validated by a vast literature and systematic reviews. Further, we used different combinations of these search words, and those were the best to retrieve the relevant literature. We have tried different methods and are confident all the literature available was retrieved.

Reviewer: L. 111-115: Were the authors able to collect all these information from the relevant literature? Whether any of the relevant meta-data was not available, would the article still be included in the meta-analyses? Additionally, sequencing technology would not be just culture-based vs sequencing, but also 454-pyrosequencing vs illumine, correct?

Response: Yes, we assessed the retrieved papers manually and extracted all relevant metadata in a spreadsheet as stated in Materials and Methods (L 105-113). Further, any paper that does not include clear metadata or was not relevant to our search was removed from subsequent meta-analysis. We initially found 151 papers and 54 papers were included in the meta-analysis, as some of the papers were reviews and/or did not contain metadata (L 104-111).

Reviewer: L. 115-119: How? Name of the sampling site? Please, clarify.

Response: Apologies, we are not sure we understand this question. Sampling sites were also extracted and reported in the metadata (see tables S1 and S2). If you are asking how some coordinates were estimated on Google earth, this was done by reading the location description section in the retrieved paper to locate the sampling site on Google Earth. Then, a middle point was taken as a representative coordinate for the study. Notably, this was done only for nine papers out of 54 papers. Once we had a complete set of coordinates for the retrieved studies, we used all coordinates to plot a representative map using ArcGIS. If several studies were conducted on the same study site/coordinate, we then used a tool named “Fishnet” on ArcGIS to plot the paper quantitively per 100 km2. I hope this is clear, but we edited the sentence to read better. This was edited and is now described as follows:

“The geographic coordinates of the sampling locations were retrieved from each study. However, for the studies that did not provide this information (n= 9), the coordinates of the sampling sites were estimated using Google Earth based on the available information in each reference. For this, coordinates were estimated by reading the location description section in the retrieved paper to locate the sampling site on Google Earth. Then, a middle point was taken as a representative coordinate for the study. Coordinates were transformed into a decimal system and plotted over the Red Sea map using ArcGIS Pro software (V 2.8.0). The Fishnet tool in ArcGIS pro (V 2.8.0) was used to quantify the number of studied sites per 100 km2 grid in the different regions of the Red Sea.”

Reviewer: L. 125-126: I know that corals are not filter feeders, but seawater microbiome would not serve as a way to determine what was specific in coral species?

Response: In fact, corals can be filter feeders (as mentioned above), and mesophotic and deep-sea corals are primarily filter feeders because of the lack of light required for photosymbiotic algae. Also, microbiomes associated with corals are host specific and distinct from the surrounding seawater and sediment (see Osman et al., 2020, 2022). Therefore, although there are some overlaps, seawater does not represent the microbiome associated with corals as well as the coral-associated microbiome per se. Therefore, it was removed from the analysis.

Reviewer: L. 128-131: Why the culture-independent and -dependent were considered separately?

Response: In this review, we assess presence/absence (i.e., occurrence) of bacterial communities associated with different coral species and regions. Therefore, the occurrence of bacteria using culture-dependent and -independent techniques would not be comparable because not all coral bacteria are culturable. Therefore, the separate approach avoids biased conclusions.

Reviewer: L. 137-138: I would strongly suggest to re-analyze the whole dataset. Considering that the results were searched from the last 20 years and bacterial taxonomy changed over the years, it does not make sense to compile an atlas which have so many distinct classifications.

Response: As stated in the Material and Methods, we used the NCBI taxonomy tool to provide a curated and up-to-date classification and nomenclature for all of the organisms in the public sequence databases. Also, NGS sequences are deposited as raw sequences in the NCBI database. NCBI aligns NGS reads against a precomputed k-mer dictionary database to provide an up-to-date taxonomy. Therefore, our data provide taxonomies independently from those provided by the authors that may vary over years because of using different versions of the SILVA/Greengenes taxonomy databases. Therefore, we are sure that we provide up-to-date taxonomy profile in our study, and also highlight that we are reporting what is currently described in the literature.

Please see more details at:

https://www.ncbi.nlm.nih.gov/sra/docs/sra-taxonomy-analysis-tool/

Reviewer: L. 139-143: I believe that articles with Sanger sequencing were involved in the culture-dependent or perhaps clone library or DGGE band, correct? How can it be compiled together with NGS, when fragments are from distinct size and 16S rRNA gene regions? Above it was stated that two datasets were analyzed (culture-independent and -dependent), how sanger and NGS combined for the atlas? Even for pipelines over time, there were distinct versions of Silva, for instance, including changes in the bacteria classification along the two decades span. There is also a sharp contrast between OTU and ASV, how can it be ignored. I cannot see the meaning of doing such a work, when so many different inputs were combined just to have a list of bacteria associated with corals. What can it add to the literature available? The authors have already all the data. Also, they have the expertise to do a meaningful analysis of the data and provide a comprehensive review of the bacteria associated with corals. Besides, why remove the duplicate, when it can provide how abundant a taxon was in a coral species and consequently their putative function?

Response: We thank the reviewer for raising these questions. We will breakdown our response below, for a better flow;

  • I believe that articles with Sanger sequencing were involved in the culture-dependent or perhaps clone library or DGGE band, correct? How can it be compiled together with NGS, when fragments are from distinct size and 16S rRNA gene regions?
  • That’s correct, Sanger sequence data was included in our analysis, but was not combined within NGS data in the analysis. Sanger sequences were assessed with the culture-dependent studies because they are not comparable to NGS, (as this would be a biased way to compare data that is not comparable). Although we agree that the reanalysis of the microbiome data potentially may provide novel insights, this approach also has several constraints, which include; i) limits our review to few papers that used Illumina sequencing technique, ii) excludes 454-pyrosequencing due to bioinformatics limitations to compare between two different sequencing technologies, and iii) neglects the culture-depended studies, which would also exclude a technique that may be popular in low-income countries surrounded the Red Sea. Together, this would bias our results toward a specific technology and ignore other important studies that have been done in the region.

Comparing/reanalyzing microbiome data from different studies that have been subjected to different library preparation, sequencing depth and protocols, sampling procedures, and many other variables may lead to even more biased results. This would produce unreliable alpha and beta diversity, which requires extensive statistical and bioinformatics validation. Most studies confirmed that reanalyzing microbiome data from different studies is inconsistent, even from within the same lab studies. Please see the following studies.

https://www.frontiersin.org/articles/10.3389/fmicb.2021.740932/full

https://www.ncbi.nlm.nih.gov/pmc/articles/PMC4523815/

https://www.ncbi.nlm.nih.gov/pmc/articles/PMC3398667/

https://bmcmicrobiol.biomedcentral.com/articles/10.1186/s12866-017-1101-8

  • It is worthwhile to note here that we used the criteria of presence/absence of bacteria in our analysis, and we did not use abundance data, which is highlighted in the text. We believe that this is the best approach to report what has been described in other studies, and avoid biased conclusions and new assignments. Because the goal of the study is to provide a catalog of microbiome diversity (i.e., an Atlas), it needs to be inclusive. So, overall, these are different methods, using different 16S regions. Yet here, we aim to remove the biases by standardizing the analysis portion and just reporting the presence of bacteria, without making potentially biased assumptions. These goals and limitations are highlighted in the new version of the manuscript.

  • Above it was stated that two datasets were analyzed (culture-independent and -dependent), how sanger and NGS combined for the atlas?
  • NGS vs Sanger sequencing data were analyzed separately, as stated in the materials and methods section. Indeed, Sanger sequencing here refers to culture-dependent, DGGE, and clone library. Such data was analyzed separately, and not combined with NGS, as mentioned. Therefore, coral microbiome data retrieved from Sanger sequence data were reported separately (Please see the result section 3.5).

  • Even for pipelines over time, there were distinct versions of Silva, for instance, including changes in the bacteria classification along the two decades span.
  • As mentioned above, we used NCBI taxonomy tool which provides curated classification and nomenclature for all of the organisms in the public sequence databases. Our data provides taxonomy independent from those provided by the authors that may vary over years because of using different versions of the SILVA/Greengenes taxonomy databases.

  • There is also a sharp contrast between OTU and ASV, how can it be ignored.
  • We agree that there is a sharp contrast between OTU and ASV, likely, among other things, due to the 97% similarity used for OTUs vs the 100% similarity of ASVs. This is one of the reasons we used only presence/absence data to report the occurrence of bacterial groups and build the bacterial atlas, based on what has been described. We agree that using abundance data obtained from OTU and ASV may cause a significant inconsistency in the results, as suggested by the reviewer, and this is why we did not reanalyze the retrieved microbiome data, but only summarized the 20 years of research as a systematic review.

  • I cannot see the meaning of doing such a work, when so many different inputs were combined just to have a list of bacteria associated with corals. What can it add to the literature available?
  • We would like to highlight that this is a review paper, not a new dataset or analysis. As a review, we summarized what is reported in the literature in a systematic way, which would otherwise be cumbersome or less readily accessible. The combined analysis of data collected, extracted, sequenced, and analyzed through different methods could also be biased and extremely tricky. We have therefore made sure to highlight that the goal is to present an overview of what is in the literature, and how the current Red Sea Atlas, based on the literature, looks like. Moreover, we believe that a distinct added value of this review is not only to summarize what is known but also to highlight what are the current gaps and/or biases. This would fill a gap in knowledge by summarizing what is known, in order to prioritize microbiome research agenda throughout the region. We have highlighted this in the MS.

  • The authors have already all the data. Also, they have the expertise to do a meaningful analysis of the data and provide a comprehensive review of the bacteria associated with corals.
  • Reanalysis of microbiome data was not the main focus of this review, and it would have several constraints, which include; i) limits our review to few papers that exclusively use Illumina sequencing technique, ii) excludes 454-pyrosequencing due to bioinformatics limitations to compare between two different sequencing technology, and iii) undervalues the culture-depended studies which remain a popular technique for low-income countries surrounded the Red Sea. This would bias our results toward specific technologies and do not address our research questions.

Second, reanalysis of different microbiome datasets that have been subjected to different library preparation, sequencing depth and chemistry, sampling time/season, and many more other variables, would produce unreliable alpha and beta diversity, which requires extensive statistical and bioinformatics validation. Most of studies confirmed that reanalyzing data from different studies are always inconsistent, even within the same lab studies. Please see the following studies.

https://www.frontiersin.org/articles/10.3389/fmicb.2021.740932/full

https://www.ncbi.nlm.nih.gov/pmc/articles/PMC4523815/

https://www.ncbi.nlm.nih.gov/pmc/articles/PMC3398667/

https://bmcmicrobiol.biomedcentral.com/articles/10.1186/s12866-017-1101-8

  • Besides, why remove the duplicate, when it can provide how abundant a taxon was in a coral species and consequently their putative function?

As mentioned above, we used presences/absence data only, and not abundance. Duplicates of same bacterial phylotypes would not be meaningful in this type of analysis.

We would like to thank the reviewers, again, for their constructive contribution. We do feel the current document better reflects our goals and methods on collecting published data that was then summarized into a catalog of the microbiome diversity (i.e., Atlas) that has been described in the Red Sea. In addition of the scientific benefits of having easy access to this data, this review also indicates the gaps and priorities that need to be tackled in order to advance microbiome research throughout the region. We are happy to address any additional questions to improve the readability and narrative of the manuscript. 

Round 2

Reviewer 3 Report

I do not have any further comment.